# Single-Cell RNA-Seq Identifies Pathways and Genes Contributing to the Hyperandrogenemia Associated with Polycystic Ovary Syndrome

**DOI:** 10.3390/ijms241310611

**Published:** 2023-06-25

**Authors:** R. Alan Harris, Jan M. McAllister, Jerome F. Strauss

**Affiliations:** 1Human Genome Sequencing Center, Baylor College of Medicine, Houston, TX 77030, USA; 2Department of Molecular and Human Genetics, Baylor College of Medicine, Houston, TX 77030, USA; 3Department of Pathology, Penn State Hershey College of Medicine, Hershey, PA 17033, USA; jxm63@psu.edu; 4Department of Obstetrics and Gynecology, Perelman School of Medicine, University of Pennsylvania, Philadelphia, PA 19104, USA

**Keywords:** polycystic ovary syndrome, single-cell RNA sequencing, human ovarian theca cells, androgens, steroids, cholesterol

## Abstract

Polycystic ovary syndrome (PCOS) is a common endocrine disorder characterized by hyperandrogenemia of ovarian thecal cell origin, resulting in anovulation/oligo-ovulation and infertility. Our previous studies established that ovarian theca cells isolated and propagated from ovaries of normal ovulatory women and women with PCOS have distinctive molecular and cellular signatures that underlie the increased androgen biosynthesis in PCOS. To evaluate differences between gene expression in single-cells from passaged cultures of theca cells from ovaries of normal ovulatory women and women with PCOS, we performed single-cell RNA sequencing (scRNA-seq). Results from these studies revealed differentially expressed pathways and genes involved in the acquisition of cholesterol, the precursor of steroid hormones, and steroidogenesis. Bulk RNA-seq and microarray studies confirmed the theca cell differential gene expression profiles. The expression profiles appear to be directed largely by increased levels or activity of the transcription factors SREBF1, which regulates genes involved in cholesterol acquisition (*LDLR*, *LIPA*, *NPC1*, *CYP11A1*, *FDX1*, and *FDXR*), and GATA6, which regulates expression of genes encoding steroidogenic enzymes (*CYP17A1*) in concert with other differentially expressed transcription factors (*SP1*, *NR5A2*). This study provides insights into the molecular mechanisms underlying the hyperandrogenemia associated with PCOS and highlights potential targets for molecular diagnosis and therapeutic intervention.

## 1. Introduction

Polycystic ovary syndrome (PCOS) is a complex endocrine disorder that affects up to 10% of women of reproductive age worldwide [1,2,3,4,5,6,7]. PCOS is characterized by various clinical features, including hyperandrogenism, irregular menstrual cycles, infertility, obesity, and insulin resistance [6,8,9,10]. Despite extensive research, the molecular mechanisms that contribute to the development of PCOS are not fully understood.

Theca cells, a component of ovarian follicles, are thought to play a crucial role in the pathogenesis of PCOS. They are responsible for producing androgens, which are converted into estrogens by granulosa cells [2,11,12]. Several studies have shown that theca cells from women with PCOS exhibit altered gene expression profiles and increased androgen production compared to theca cells from women without PCOS [7,11,12,13,14,15,16,17,18,19,20]. However, the molecular mechanisms that underlie these changes in theca cell function have not been elucidated.

Single-cell RNA sequencing (scRNA-seq) [21] is a powerful tool for identifying molecular pathways and genes that are affected by the disease. This technology allows for identifying cell subpopulations, gene expression patterns, and gene regulatory networks at high resolution. In the context of PCOS, scRNA-seq can be used to identify molecular changes in theca cells that are associated with disease pathogenesis, as well as to identify potential targets for therapeutic intervention.

Here, we describe using scRNA-seq to identify pathways and genes affected by PCOS in theca cells. Our findings provide new insights into the molecular mechanisms underlying PCOS, which may have diagnostic and therapeutic implications for this common endocrine disorder.

## 2. Results

### 2.1. scRNA-Seq Reveals That PCOS Theca Cells Have Distinctive Gene Expression Profiles

The 10× Cell Ranger [21] processing of the raw scRNA-seq data identified 70,223 cells across all 20 samples from women with and without PCOS that were untreated or forskolin-treated. After standard filtering for mitochondrial read counts < 5% and a number of gene features > 200 using Seurat [22], a total of 20,775 cells (29.58% of total cells) were retained. After integration with sub-type anchoring correction for alignment in Seurat (STACAS) [23] and additional filtering consisting of removing outlier cells based on UMAP visualization, a total of 15,565 (22.17%) theca cells were included in the downstream analyses. After all filtering, the number of cells per group was: unaffected/untreated 4672; unaffected/forskolin-treated 4793; PCOS/untreated 3446; PCOS/forskolin-treated 2654. UMAP visualizations of the filtered cells show the clustering of groups by affection status/treatment (Figure 1a), which is also consistent with the clustering of the samples in each group (Figure 1b).

Gene set enrichment analysis (GSEA) of molecular signatures database (MSigDB) C2: curated gene sets [24] using singleseqgset [25] identified pathways significantly (FDR corrected *p* < 0.05) differentially expressed in comparisons across the four groups. Among these significant pathways were ones involved in cholesterol acquisition (six with CHOLESTEROL in the pathway terms) and steroidogenesis (10 with ANDROGEN or STEROID in the pathway terms) (Figure 2 and Appendix A).

Working from the significant differentially expressed pathways, we identified individual genes in the pathways that were themselves significantly (Bonferroni corrected *p* < 0.05) differentially expressed genes (DEGs). We identified 168 genes differentially expressed in at least one group, including 18 genes in cholesterol acquisition and steroidogenesis pathways. There were 59 DEGs involved in cholesterol acquisition (Appendix A). These included the gene encoding the rate-limiting enzyme in de novo cholesterol biosynthesis (*HMGCR*) and the transcription factor *SREBF1*, which regulates genes involved in cholesterol acquisition.

We identified 127 DEGs involved in steroidogenesis (Appendix A). These included *STAR*, which catalyzes the rate-limiting step in steroid synthesis, the translocation of cholesterol to the inner mitochondrial membrane where the cholesterol side-chain cleavage enzyme, CYP11A1, another DEG, resides. The genes encoding the CYP11A1 electron transport proteins (*FDX1*, *FDXR*), the enzyme involved in the metabolism of pregnenolone to DHEA (*CYP17A1*), and proteins that enhance the CYP17A1 17/20-lyase reaction (*POR*, *CYB5A*) [26,27,28], were also differentially expressed, with highest expression in PCOS theca cells stimulated with forskolin.

Based on transcription factors identified by Lambert et al. [29], the cholesterol acquisition pathways had six differentially expressed transcription factors (*NCOA1*, *NR1H2*, *NR1H3*, *RXRA*, *RXRB*, and *SREBF1*). The steroidogenesis pathways had 15 differentially expressed transcription factors (*AR*, *CREB1*, *FOXO1*, *JUN*, *KAT7*, *MAF*, *MYC*, *NCOA1*, *NCOA3*, *NR2C2*, *NR3C1*, *PATZ1*, *SP1*, *ZBTB10*, and *ZNF318*).

Among the individual DEGs in significant differentially expressed pathways, a number are genes that have been previously suggested as having a role in PCOS. There are 18 significant genes in cholesterol acquisition pathways (Appendix A) and 31 significant genes in steroidogenesis pathways that are potential drivers of the thecal PCOS phenotype (Appendix A).

In addition to genes previously suggested to be involved in producing the hyperandrogenemia phenotype, we identified several other DEGs that could potentially be involved in the pathogenesis of PCOS. There were 41 potential novel candidates in cholesterol acquisition pathways (Appendix A) and 96 potential novel candidates in steroidogenesis pathways (Appendix A).

We also examined significant DEGs not annotated in these pathways but likely to play a role in those functions (Appendix A). The cholesterol acquisition genes *LDLR*, *LIPA*, *NPC1*, and *SCARB1* were significantly (Bonferroni corrected *p* < 0.05) differentially expressed. They are involved in the uptake of and processing of lipoprotein-carried cholesterol. Low-density lipoprotein cholesteryl esters, taken into cells by LDLR, are hydrolyzed by LIPA in lysosomes, and free cholesterol is released from lysosomes by NPC1 and then trafficked to various subcellular compartments, including the mitochondria where the first committed step in steroid hormone synthesis takes place, cholesterol side-chain cleavage [30,31]. The steroidogenesis-associated genes encoding transcription factors *GATA4*, *GATA6*, *NR5A1*, and *NR5A2* were significantly (Bonferroni corrected *p* < 0.05) differentially expressed, as was *CITED2*.

In Figure 3, we highlight a subset of significant DEGs either in differentially expressed pathways or among the additional genes we added that are likely to play a major role in cholesterol acquisition and steroidogenesis. These genes are regulated by two transcription factors that control genes involved in cholesterol acquisition and metabolism (SREBF1) and steroidogenesis (GATA6), both of which were found to be elevated in PCOS theca cells.

In order to examine associations among proteins translated from DEGs identified by scRNA-seq analysis, we use the STRING [32] protein–protein interaction database. Figure 4 is a STRING interaction network visualization of the 13 proteins translated from DEGs under the header “Cholesterol acquisition” in the Figure 3 heatmap and the transcription factor SREBF1. SREBF1 is directly associated with 10 of the cholesterol acquisition proteins, which excludes APOC1, ACAT1, and ACAT2. HMGCR, the rate-limiting enzyme in de novo cholesterol biosynthesis, is directly associated with all the proteins in the network except for APOC1.

Figure 5 is a STRING interaction network visualization of the 16 proteins translated from scRNA-seq identified DEGs under the header “Steroidogenesis” in the Figure 3 heatmap together with the transcription factor GATA6. GATA6 shows direct associations with CYP17A1 and HSD17B1. CYP17A1 is directly associated with all the proteins in the network except for LIPA, INSIG1, FDX1L STARD4, and STARD3NL. FDX1L is indirectly associated with CYP17A1 through FDXR and FDX1. STARD4 and STARD3NL are indirectly associated with CYP17A1 through STAR.

### 2.2. Bulk RNA-Seq Confirms a Distinctive Repertoire of Gene Expression in PCOS Theca Cells

We performed bulk RNA-seq on theca cells from normal and PCOS women treated with forskolin. In contrast to scRNA-seq, which characterizes the transcriptomes of individual cells, bulk RNA-seq assesses average gene expression across a large population of cells. We then used Cufflinks [33] to identify significant (*q* value < 0.05) DEGs and compared the bulk RNA-seq DEGs to scRNA-seq DEGs from pairwise comparisons of normal and PCOS theca cells treated with forskolin (Appendix A). In this comparison, 86 significant DEGs were shared between scRNA-seq and bulk RNA-seq and showed fold changes in the same direction, including *GATA6* and *CYP17A1.* The findings confirmed the differential expression of genes involved in cholesterol acquisition and androgen biosynthesis.

### 2.3. Differential Gene Expression Identified by scRNA-Seq Is Consistent with the PCOS Molecular Signature Detected by Microarrays

We compared DEGs discovered in our scRNA-seq dataset to our previously published microarray data obtained from the analysis of normal and PCOS theca cells cultured without and with forskolin [34]. In forskolin-treated theca cells, 143 DEGs were shared between scRNA-seq and microarray datasets and showed fold changes in the same direction (Appendix A). A total of 15 DEGs were shared across scRNA-seq, microarray, and bulk RNA-seq datasets with fold changes in the same direction, including GATA6 and *CYP17A1*. There were 128 PCOS candidates shared only between scRNA-seq and microarray from forskolin-treated theca cells, including *HMGCR*, *FDX1*, *CYB5A*, and *CITED2*. In untreated theca cells, 100 DEGs were shared between scRNA-seq and microarray datasets and showed fold changes in the same direction, including the PCOS candidate STAR (Appendix A).

## 3. Discussion

Our study provides new insights into the molecular phenotype of PCOS theca cells. Our findings suggest that theca cells from women with PCOS have an enhanced capacity to acquire cholesterol and metabolize it into androgens, which disrupt the function of the ovarian–hypothalamic–pituitary axis, leading to anovulation and infertility, as well as promoting metabolic dysfunction. In addition, our study identified several novel candidate genes that may be involved in the pathogenesis of PCOS. Future studies are needed to validate these findings and to elucidate the functional roles of these genes in PCOS. Of note, genes involved in the acquisition of cholesterol, the precursor of ovarian steroid hormones, including those involved in de novo biosynthesis or uptake of lipoprotein-carried cholesterol, were up-regulated in PCOS theca cells, especially when they were stimulated with forskolin. Many of these genes are transcribed in response to the activation of the transcription factor SREBF1. Additionally, the expression of genes encoding steroidogenic proteins catalyzing the metabolism of cholesterol into pregnenolone and subsequently into androgens was elevated in PCOS theca cells [12,13,35]. Of note, genes encoding proteins that increase the 17/20-lyase activity of CYP17A1, which converts pregnenolone into DHEA (*POR*, *CYB5A*), were significantly up-regulated.

Because cholesterol is the precursor of steroid hormones, there is an expected overlap among the proteins depicted in the STRING interaction networks presented in Figure 4 (Cholesterol acquisition) and Figure 5 (Steroidogenesis). For example, *LIPA* encodes the lysosomal acid lipase involved in the hydrolysis of LDL-carried cholesteryl esters, thus generating free cholesterol that can be trafficked to the mitochondria, where the first committed step in steroidogenesis takes place. Mutations inactivating *LIPA* result in lysosomal accumulation of cholesteryl esters (Wolman disease), which damages steroidogenic cells, reducing steroidogenesis [36,37].

Transcription factors controlling theca cell cholesterol acquisition (SREBF1) [38] and androgen biosynthesis (GATA6) were significantly increased in PCOS theca cells [39,40], especially when they were stimulated with forskolin. Notably, statin drugs, which inhibit the HMGCR protein product 3-hydroxy-3-methyl-glutaryl-CoA (HMG-CoA) reductase, the rate-limiting enzyme in cholesterol biosynthesis, impact the growth of human theca cells and reduce androgen levels in women with PCOS, consistent with a key role for cholesterol metabolism in the theca cell dysfunction in PCOS [41]. In addition to GATA6 and SREBF1, other transcription factors are involved in differential gene expression. For example, GATA6 activates *CYP17A1* transcription by binding to another differentially expressed transcription factor, SP1, that interacts with the *CYP17A1* proximal promoter [42,43].

It should be noted that differential gene expression detected by RNA-seq or micro-arrays can reflect differences in gene transcription as well as differences in mRNA half-life. For example, we have previously shown that in addition to increased transcription, *GATA6* [40], *CYP11A1* [44], and *CYP17A1* [45] gene expression is augmented in PCOS theca cells by increased mRNA stability. Thus, transcriptional and post-transcriptional mechanisms contribute to the increased transcript levels encoded by these genes in PCOS theca cells.

We have employed three different methods (scRNA-seq, bulk RNA-seq, and microarray methods) to analyze the molecular phenotype of normal and PCOS theca cells. Overall, the findings are congruent with increased expression of genes encoding proteins involved in cholesterol biosynthesis or uptake and metabolism to androgens. Each of the methods we have employed has strengths and limitations. For example, a limitation of the scRNA-seq approach to profiling gene expression is that most scRNA-seq methods identify DEGs at the gene level but not at the isoform level due to the limitations of current single-cell sequencing technologies and software analysis tools [46]. Consequently, the scRNA-seq method we employed could fail to detect variation in the expression of some isoforms that play important pathophysiologic roles in PCOS. Microarray studies can fail to detect expressed genes not represented in the nucleotide arrays, in addition to failing to detect splice variants if the arrayed nucleotide sequences are not complementary to sequences in the variant transcripts. Bulk RNA-seq, while capable of identifying splice variants, is less sensitive and may fail to detect variants present in low abundance or not represented in genomic databases [47].

As an example, we previously demonstrated that DENND1A variant transcript 2 (DENND1A.V2) is elevated in PCOS theca cells (by qPCR and western blotting). That forced expression of DENND1A.V2 in normal theca cells augments androgen production, and conversely, that DENND1A.V2 mRNA knock-down in PCOS theca cells diminishes androgen production [7]. However, DENND1A.V2 was not identified as significantly differentially expressed in our scRNA-seq or microarray studies but was increased in bulk RNA-seq experiments, although the increase was not statistically significant. We attribute the failure to detect differential expression of this variant transcript, which encodes a major PCOS candidate gene, to the limitations of the abovementioned methods.

DENND1A.V2 encodes a protein associated with coated pits, where gonadotropin receptors cluster to initiate signal transduction followed by endocytosis and vesicular trafficking through the cytoplasm [7]. The DENND1A.V2 traffic pattern includes translocation into the nucleus, where we have proposed that it plays a role in initiating the transcription of genes involved in androgen biosynthesis [7,48]. Our scRNA-seq findings suggest that DENND1A.V2 could trigger the expression of SREBF1 and GATA6, which in turn, could account for the DENND1A.V2-driven increase in theca cell androgen production through transcriptional activation of genes encoding steroidogenic enzymes.

## 4. Materials and Methods

### 4.1. Theca Cell Cultures

Human theca interna tissue was obtained from follicles of women undergoing hysterectomy, following informed consent under a protocol approved by the Institutional Review Board of The Pennsylvania State University College of Medicine. As a standard of care, oophorectomies were performed during the luteal phase of the cycle. Theca cells from normal cycling and PCOS follicles were isolated and grown as previously reported in detail [7,20,40,49,50]. PCOS and normal ovarian tissue came from age-matched women, 38–41 years old. The diagnosis of PCOS was made according to the National Institutes of Health (NIH) consensus guidelines [9,10], which include hyperandrogenemia/hyperandrogenism and oligo-ovulation and the exclusion of other causes of hyperandrogenemia (e.g., 21-hydroxylase deficiency, Cushing’s syndrome, and adrenal or ovarian tumors). All PCOS theca cell preparations studied came from the ovaries of women with fewer than six menses per year and elevated serum total testosterone or bioavailable testosterone levels [11,13,14]. Each PCOS ovary contained multiple subcortical follicles less than 10 mm in diameter. The control (normal) theca cell preparations came from ovaries of fertile women with normal menstrual histories, menstrual cycles of 21–35 days, and no clinical signs of hyperandrogenism. Neither PCOS nor normal subjects were receiving hormonal medications at the time of surgery. Indications for surgery were dysfunctional uterine bleeding, endometrial cancer, and pelvic pain. Experiments comparing PCOS and normal theca were performed using fourth-passage (31–38 population doublings) theca cells isolated from individual size-matched follicles obtained from age-matched subjects in the absence of in vivo stimulation. The use of fourth-passage cells allowed us to perform multiple experiments from the same patient population that were propagated from frozen stocks of second-passage cells in the media described above. The passage conditions and split ratios for all normal and PCOS cells were identical. These studies were approved by the Human Subjects Protection Offices of Virginia Commonwealth University and Penn State College of Medicine.

### 4.2. Single-Cell RNA Sequencing (scRNA-Seq)

For scRNA-seq studies, fourth passage theca cells were gown and propagated from five theca cell preparations obtained from women with PCOS (MC03, MC10, MC16, MC26, and MC27), and five theca cell preparations from normal cycling women (MC02, MC06, MC31, MC40, and MC50), in a serum-containing medium, in a 5% O_2_/5% CO_2_ incubator, as we have previously described [7,20,40,49,50]. All of the subjects we recruited were unrelated and of European ancestry. Once all of theca cell preparations were approximately 85–95% confluent, the cells were rinsed and transferred to serum-free defined media and treated under basal control conditions or treated with forskolin (20 μM), an activator of adenylate cyclase for 24 h. Following treatment, the cells were rinsed with 5mL of PBS and trypsinized and harvested in a cold 1:1 DME/F12 medium containing 10% FBS, as we have previously described [7,20,40,49,50]. The cells were then centrifuged at 600× *g*, and the cells were resuspended according to Active Motif’s (Carlsbad, CA, USA) protocol in 20% DMSO and 50% FBS in 1:1 DME/F12 and frozen in 2 mL cryovials using a Corning isopropanol −80 degree cryofreezing apparatus for 24 h. Afterward, the cells were transferred and stored in a ThermoFisher liquid N_2_ cell storage unit (Waltham, MA, USA) until all of the theca cell preparations were processed and frozen in lq N2 stable cryovials, prior to sending the whole frozen cell preparation to Active Motif (Carlsbad, CA, USA) for subsequent cell dilution, RNA isolation, and single-cell library preparation, using Active Motifs proprietary conditions. Following single-cell library preparation, 10× single-cell RNA sequencing (scRNA-seq) was performed using an Illumina NextSeq 500 (San Diego, CA, USA) sequencing apparatus to generate 91 bp sequencing reads.

### 4.3. Bulk RNA Sequencing (Bulk RNA-Seq)

For bulk RNA-seq experiments, fourth passage theca cells were gown and propagated from four theca cell preparations obtained from women with PCOS (MC03, MC10, MC26, and MC27) and four theca cell preparations from normal cycling women (MC02, MC31, MC40, and MC50), in a serum-containing medium, in a 5% O_2_ and 5% N_2_ incubator, as we have previously described [7,20,40,49,50]. As described above for scRNA-seq, once the cells were 85–95% confluent, all of the cells were rinsed with PBS, and transferred into a serum-free medium in the presence of 20 mM forskolin for 16 h. Following treatment, the cells were rinsed in PBS, and after complete aspiration, the cell dishes were tightly wrapped in foil, labeled, and flash frozen in Liq N_2_ prior to storage in a −80 degree ThermoFisher freezer (Waltham, MA, USA). Once all the cells were treated and collected, we harvested RNA using Sigma Tri-reagent, using Sigma’s standard protocol for cells in monolayer culture, as we have previously described [7,20,40,49,50], following isopropanol precipitation and centrifugation at 10× *g* in a 4 degrees C microfuge for 30 min. The resulting total RNA cell pellet was rinsed in 70% ethanol and re-centrifuged at 10× *g* in a Beckman 4 degrees C microfuge for 20 min. The resulting total RNA pellet was resuspended in 65 mL RNAase-free ultrapure water, OD’d, and aliquoted into 5–10 mL aliquots, then frozen in a −80 freezer prior to transfer to ThermoScientific (previously Life Sciences) (Waltham, MA, USA) for library preparation for bulk RNA-seq. The bulk RNA-seq was performed by sequencing paired-end 100 bp reads at a depth of 100 million reads using Illumina HiSeq (San Diego, CA, USA).

### 4.4. Cell Ranger Data Processing

Cell Ranger [21] v.6.1.1 was used to process the raw scRNA-Seq data. The Cell Ranger mkfastq function was run with default parameters to demultiplex raw base call (BCL) files generated by Illumina sequencers into FASTQ files. The Cell Ranger count function was run with default parameters to perform genome alignment, filtering, barcode counting, and unique molecular identifier (UMI) counting that assigns each read to its originating cell and transcript. The transcript counts were then aggregated to generate a matrix of gene expression values for each cell.

### 4.5. Seurat Pre-Processing and Quality Control

The Seurat [22] v.4.1.1 package loaded into R v.4.1.3 performed standard pre-processing and quality control on the scRNA-Seq data. For each sample, Cell Ranger output files were read using Read10×. The percent of mitochondrial genes was calculated, and cells with >5% mitochondrial genes were removed. Cells with <200 RNA features were also removed. The Seurat functions NormalizeData and FindVariableFeatures were then run using default parameters.

### 4.6. STACAS Integration

The Sub-Type Anchoring Correction for Alignment in Seurat (STACAS v.2.0.1) [23] package loaded into R v.4.1.3 was used to perform semi-supervised integration of the individual theca cell scRNA-seq datasets. Highly variable genes (HVG) were identified, excluding blocklist genes consisting of mitochondrial, ribosomal, heat-shock, and cell cycle genes. PCOS candidate genes and other genes known to be expressed in the ovary were selected from the HVG genes as features for integration. Integration was performed using the Run.STACAS function with default parameters except for anchor.features set to the ovarian genes, cell.labels set to the known affection status and treatment of the cell, and label.confidence set to 0.99. Setting the cell.labels parameter causes STACAS to perform semi-supervised integration by using this information to penalize anchors where cell types are inconsistent.

### 4.7. UMAP Generation

Uniform Manifold Approximation and Projection (UMAP) visualizations were generated using the Seurat RunUMAP function with default parameters except for n.neighbors = 50, min.dist = 0.5 to optimize for global structure. The initial UMAP visualization was used to identify 5210 outlier cells that were removed.

### 4.8. Identification of Differentially Expressed Genes

The Seurat FindMarkers function was used to identify differentially expressed genes across the groups based on known affection status and treatment of the cell. FindMarkers was run with default parameters except for min.pct = 0.01 and logfc.threshold = 0.01 to allow for more sensitive detection of differential expression. This function calculates a Wilcoxon rank-sum test for each gene to compare its expression level in each individual group to all the other groups combined. The test results were then adjusted for multiple testing using Bonferroni correction.

### 4.9. Gene Set Enrichment Analysis

The singleseqgset [25] v.0.1.2.9000 package loaded into R v.4.1.3 was used to perform Gene Set Enrichment Analysis (GSEA). The logFC function was used to calculate the log fold change between each group and all the other groups combined for all genes using log-normalized data to correct for library size. The gse.res function was used to calculate enrichment scores and perform Wilcoxon rank sum tests, followed by Benjamini–Hochberg FDR correction, on the gene sets. The gene sets used were the Molecular Signatures Database (MSigDB) C2: curated gene sets [24], which are curated from sources, including online pathway databases, biomedical literature, and contributions by individual domain experts.

### 4.10. STRING Protein–Protein Interaction Database and Network Visualization

The STRING [32] protein–protein interaction database was used to identify associations between proteins translated from differentially expressed genes. Protein–protein interaction networks were generated and visualized using the Cytoscape stringApp plugin [51].

### 4.11. Bulk RNA Sequencing Analysis

Tophat [33] v.1.4.1 was used to align RNA-seq reads to the human (GRCh37/hg19) genome assembly. Cufflinks [33] v.2.1.1 was used to perform differential expression analysis.

## 5. Conclusions

Our study demonstrates the feasibility and utility of performing scRNA-seq on theca cells to identify molecular pathways and genes affected by PCOS. Differentially expressed genes in PCOS theca cells included those involved in the acquisition of cholesterol, the precursor of steroid hormones, and genes involved in the biosynthesis of androgens. The transcription factors SREBF1, which regulates genes involved in cholesterol acquisition, and GATA6, which regulates the expression of genes encoding steroidogenic enzymes (CYP17A1, CYP11A1) in concert with other differentially expressed transcription factors, appear to play a major role in driving the PCOS theca cell molecular signature. Our findings provide new insights into the molecular mechanisms underlying theca cell dysfunction in PCOS and identify potential targets for therapeutic intervention. scRNA-seq of theca cells may also have diagnostic and prognostic applications for PCOS.

## Figures and Tables

**Figure 1 ijms-24-10611-f001:**
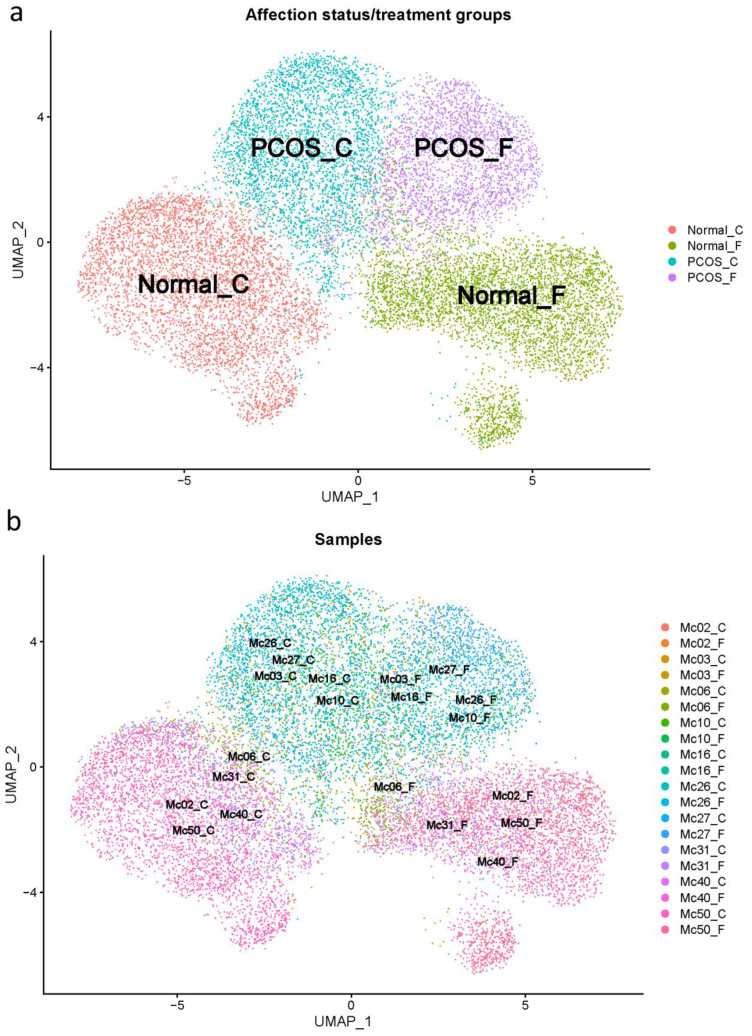
UMAP of affection status/treatment groups (**a**) and samples (**b**). Values from normal (control) theca cells were generated using the five theca preparations from PCOS women (MC03, Mc10, MC16, MC26, and MC27) and five theca cell preparations from normal cycling women (MC02, MC06, MC31, MC40, and MC50), that were grown until subconfluent, and treated with (F) and without (C), 20 mM forskolin for 24 h in a serum-free medium.

**Figure 2 ijms-24-10611-f002:**
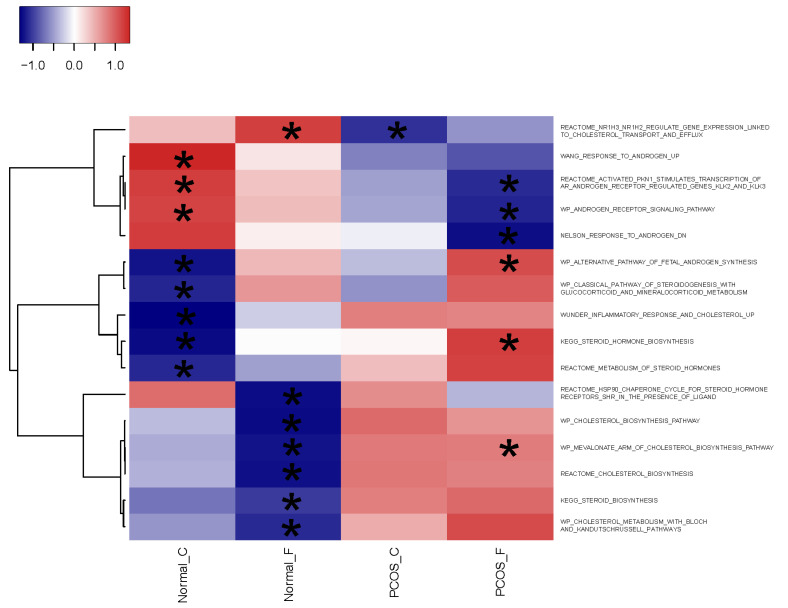
Heatmap of the Z scores for cholesterol acquisition and steroidogenesis pathways from molecular signatures database (MSigDB) C2: curated gene sets significantly enriched in at least one group. Asterisks show the significant differentially expressed group(s). scRNA-seq data were generated using theca cell preparations from normal cycling and PCOS women grown until subconfluent and either untreated (C) or treated (F) with 20 μM forskolin for 24 h in a serum-free medium.

**Figure 3 ijms-24-10611-f003:**
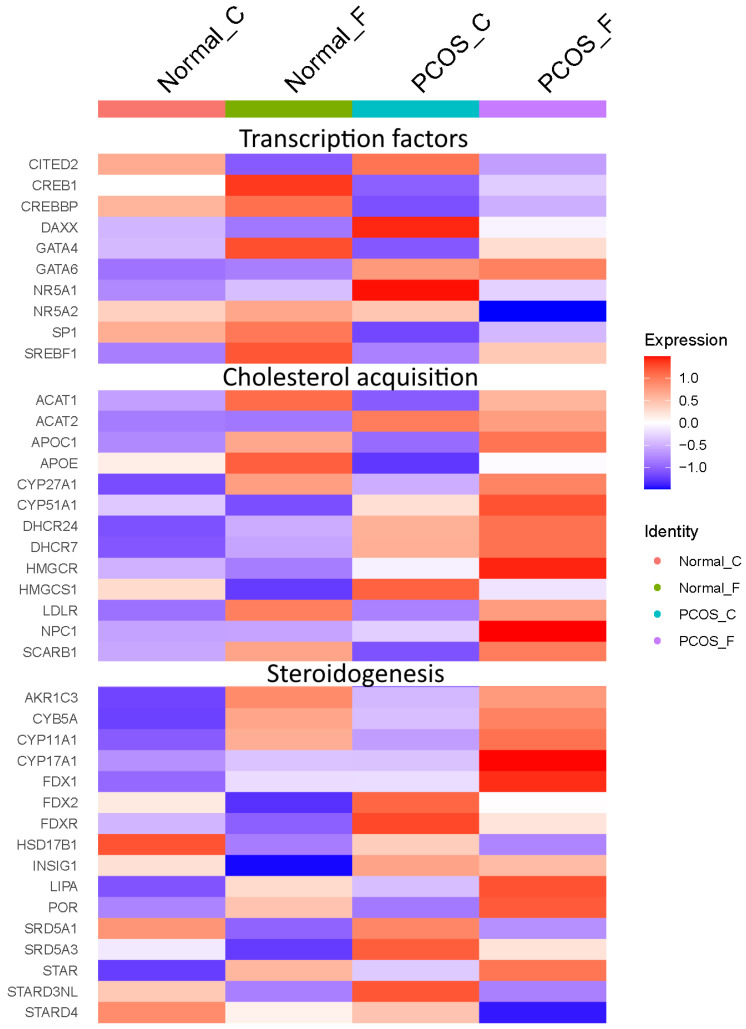
Heatmap of significant differentially expressed genes likely to play a major role in cholesterol acquisition and steroidogenesis.

**Figure 4 ijms-24-10611-f004:**
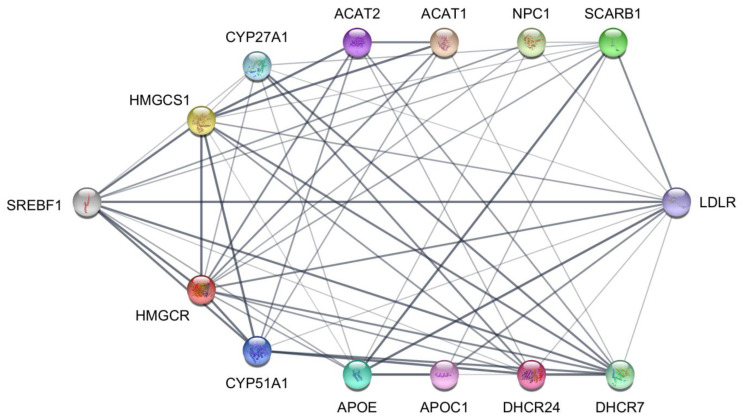
STRING protein–protein interaction network of cholesterol acquisition proteins translated from differentially expressed genes identified by scRNA-seq of normal and PCOS theca cells. The nodes represent the proteins, and the width of the edges shows the confidence of the association between connected proteins.

**Figure 5 ijms-24-10611-f005:**
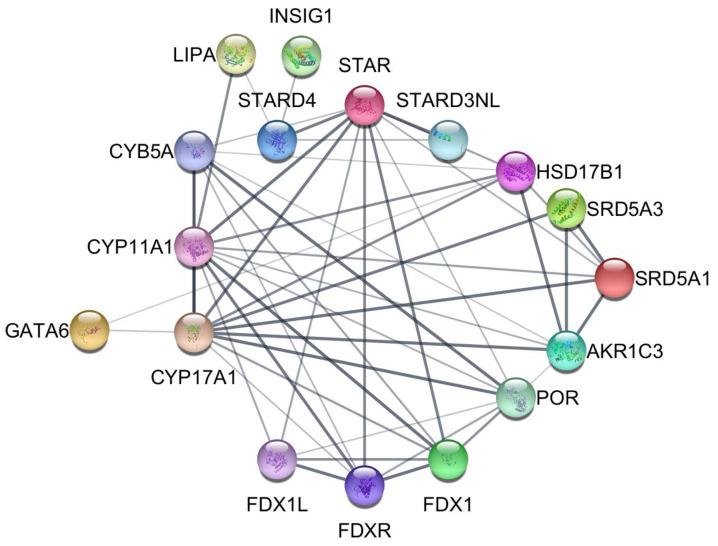
STRING protein–protein interaction network of steroidogenesis proteins translated from significant differentially expressed genes identified by scRNA-seq of normal and PCOS theca cells. The nodes represent the proteins, and the width of the edges shows the confidence of the association between connected proteins.

## Data Availability

The 10× Cell Ranger filtered feature-barcode matrix data files and the scRNA-seq data are publicly available for download at https://zenodo.org/record/7942968 (accessed on 21 May 2023). These files are the read counts by gene by cell that can be used to analyze single cell data.

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
