# Peer review of "Single-Cell RNA-Seq Identifies Pathways and Genes Contributing to the Hyperandrogenemia Associated with Polycystic Ovary Syndrome"

_ijms, 2023, doi:10.3390/ijms241310611_

Round 1

Reviewer 1 Report

This is a very well written and well illustrated paper describing the results of single cell RNAseq analysis of cultured theca cells from women with and without PCOS. The paper focuses on steroidogenic pathways and how they differ between PCOS and control samples.

I have a few minor comments:

1.     The authors mention that one of the advantages of scRNAseq over global RNAseq is to define subpopulations of cells within groups (as well as between groups) but this aspect does not seem to appear in their results and discussion. It would be helpful if the authors could enlarge on this.

2.     It is curious that differential expression of DENDD1A was not a feature of the results of scRNAseq, as might have been expected. Is this, as the authors suggest, a possible limitation of this approach or could there be another explanation? For example, it is not clear whether the passaged theca cell population that was used in this study was from the same primary population as in their previous study. Could that be a factor? 

3.     It would be interesting to have the authors’ views on whether they would expect different results from freshly harvested theca cells either before or after primary culture.

Author Response

See uploaded ijms-2455062.reply.to.reviewers.reviewer.1.docx which may be renamed by the submission system.

Reviewer 2 Report

The article by Harris et al. puts forward some interesting facts. The manuscript is well-planned and structured. However, I suggest some modifications before the publication of the article.

The abstract should be structurally divided into brief background, methodology, significant findings, results, and conclusions. It will make it more easy and interesting for the readers.

The introduction needs a major modification. The references cited in the introduction section are outdated. Therefore more recent articles need to be referred to, and a clear background should be provided along with the study's objective.

The link between the three keywords PCOS, theca cells, and scRNA-seq should be linked and described. Only a small paragraph defining the terms is not enough.

Several sections in the methodology section lack sufficient data and look incomplete. These sections should be carefully revised, and the necessary information should be added.

The discussion section needs to be elaborated more, and the importance of the study should be addressed.

The conclusion section has been written in a hurry and should be included with author's views and perspectives.

Some typos and grammatical errors need to be addressed.

Minor editing of English language required

Author Response

Please see uploaded ijms-2455062.reply.to.reviewers.reviewer.2.docx which may be renamed by the submission system.

Reviewer 3 Report

The manuscript by Harris et al. carries out an exhaustive characterization of genes involved in human theca cell function, comparing between cells of control and PCOS patients. For this purpose, the authors employed three different methods (scRNA-seq, bulk RNA-seq, and microarray methods). Results are clear, comprehensive, and orderly presented, and provide answers to some of the unresolved questions in this topic. The discussion is relevant, comprehensive, and up-to-date. The latest advances in the subject have been considered.

Minor comments:

Page 9, lines 243-244: Review the wording (“…and treated the cells were treated under basal control conditions,…)

In Supplementary Materials - Table S4 – Row 1 - Change "Table S3" to "Table S4"

Author Response

Please see uploaded ijms-2455062.reply.to.reviewers.reviewer.3.docx which may be renamed by the submission system.

Round 2

Reviewer 2 Report

The modified version of the manuscript is fine and can be accepted for publication.